# The role of charge in microdroplet redox chemistry

**Joseph P. Heindel[1,2], R. Allen LaCour[1,2] & Teresa Head-Gordon** [1,2,3] ✉

In charged water microdroplets, which occur in nature or in the lab upon ultrasonication or in electrospray processes, the thermodynamics for reactive chemistry can be dramatically altered relative to the bulk phase. Here, we provide a theoretical basis for the observation of accelerated chemistry by simulating water droplets of increasing charge imbalance to create redox agents such as hydroxyl and hydrogen radicals and solvated electrons. We compute the hydration enthalpy of $OH^-$ and $H^+$ that controls the electron transfer process, and the corresponding changes in vertical ionization energy and vertical electron affinity of the ions, to create $OH^{\cdot}$ and $H^{\cdot}$ reactive species. We find that at ~20−50% of the Rayleigh limit of droplet charge the hydration enthalpy of both $OH^-$ and $H^+$ have decreased by >50 kcal/mol such that electron transfer becomes thermodynamically favorable, in correspondence with the more favorable vertical electron affinity of $H^+$ and the lowered vertical ionization energy of $OH^-$. We provide scaling arguments that show that the nanoscale calculations and conclusions extend to the experimental microdroplet length scale. The relevance of the droplet charge for chemical reactivity is illustrated for the formation of $H_2O_2$, and has clear implications for other redox reactions observed to occur with enhanced rates in microdroplets.

Recent experimental work has demonstrated that many reactions are accelerated by one to six orders of magnitude in aqueous microdroplets when compared to bulk liquid water[1–7]. Of particular interest are the many observations of redox chemistry occurring at accelerated rates[6,8–12], as well as the observation of hydrogen peroxide ($H_2O_2$) formation in microdroplets at concentrations of ≈1−2 μM[13,14]. Since the production of $H_2O_2$ in water is thermodynamically disfavored, these experiments collectively point to microdroplets having strong redox properties. At present, however, the molecular origins of the strong redox properties of microdroplets are unclear.

Microdroplets have inspired much theoretical work to explain the observed increase in reactivity by focusing on the defining features of a microdroplet that distinguish it from the bulk liquid[15–18], such as the large surface area to volume ratio and the unique structural and dynamical aspects of the droplet surface. One possible explanation is that molecules at the droplet surface are partially solvated relative to

the bulk, which destabilizes the reactants and/or decreases the reaction barrier[4,10,19]. Indeed, we recently showed that partially solvated $OH^-$ ions have much lower vertical ionization energies (VIEs)[18] to yield the $OH^{\cdot}$ radical, and that these under-coordinated $OH^-$ ions are much more likely to be within 1 nm of the air–water interface[18]. Previous studies have suggested that HO-radicals, and consequently $H_2O_2$, can be formed at the surface because partially hydrated $HO^-$ and $H^+$ ions undergo electron transfer during encounters between charged microdroplets[20], and/or because of the high electric field generated by contact electrification[21].

A related possibility is rate acceleration arising from the presence of intrinsic electric fields at the air-water interface[17,18]. As shown by Hao and co-workers[17], the average electric fields at the interface are not large, but the interface is different from the bulk liquid by exhibiting Lorentzian electric field distributions due to fluctuations. The long tails of the electric field distribution give a finite probability for large

---

[1]Kenneth S. Pitzer Theory Center and Department of Chemistry, Berkeley, CA, USA. [2]Chemical Sciences Division, Lawrence Berkeley National Laboratory, Berkeley, CA, USA. [3]Departments of Bioengineering and Chemical and Biomolecular Engineering University of CAlifornia, Berkeley, CA, USA. ✉e-mail: thg@berkeley.edu

electric field events[17] that can reach the requisite energy scales for chemical reactivity, such as ionizing an electron from the hydroxide ion[18]. The work of Hao is supported by experimental measurements of surface electric fields inferred by the Stark effect[22].

We note that although there is much heated experimental debate about microdroplet reactivity at the air-water interface[13,23], no experiment has directly determined the fate of the electron, even though the knowledge of its fate is a critical requirement of any proposed redox chemistry mechanism. Very recently, however, $H_2$ gas has been observed to be produced by oil−water emulsions along with observation of H[·] and OH[·] radicals using electron paramagnetic resonance[24]. These observations provide important additional evidence that water-oil interfaces can drive redox chemistry. There is some debate about the similarity of oil−water emulsions and the air-water interface[25–27], but the work of Chen[24] confirms that redox chemistry can certainly occur at oil-water interfaces formed by emulsification.

Another important aspect of microdroplet chemistry is that in the preparation of droplets using ultrasonication, electrospray, as well as gas nebulization, microdroplets are formed with a net charge[14]. There is very good evidence that charged droplets occur not only in the laboratory[14,28] but in nature as well[29–31], and there are consistent observations that smaller droplets tend to have a more negative charge while larger droplets tend to have a more positive charge when produced via natural processes[21,29,30,32]. Furthermore, a microdroplet with a net charge is not the same as an ionic solution which always contains counterions. While ionic solutions can modulate the properties and reaction thermodynamics compared to neutral solutions[33–35], the net charge carried by a microdroplet offers the possibility of a more dramatic alteration of the reaction thermodynamics. This motivates us to examine redox reaction activity in a charged microdroplet as an origin of difference relative to a bulk ionic system.

Redox chemistry requires an oxidizing agent and/or a reducing agent. Considering most microdroplet chemistry is done in liquid water, the most likely redox agents are hydroxyl radicals (oxidation) and solvated electrons (reduction):

$$OH^-(aq) \rightleftharpoons OH^{\bullet}(aq) + e^-(aq) \tag{1}$$

Ordinarily, the equilibrium constant associated with Eq. (1) is negligibly small since $OH^-$ is more strongly solvated than the electron[36,37], which is why $H_2O_2$ is not normally produced in bulk $H_2O$. The solvated electron is highly reactive, so if it were produced it would rapidly reduce any available species such as an organic molecule or even protons produced by water autoionization, beyond the recombination with OH[·] via the backwards reaction in Eq. (1).

$$H^+(aq) + e^-(aq) \rightleftharpoons H^{\bullet}(aq) \tag{2}$$

More quantitively, the process formed by combining the half-reactions in Eqs. (1) and (2) is,

$$OH^- + H^+ \rightarrow OH^{\bullet} + H^{\bullet}, \Delta H = 107\ kcal/mol \tag{3}$$

While the hydration entropy is only minorly affected by excess charges in the environment, the process described in Eq. (3) is strongly disfavored by the enthalpy under neutral bulk conditions as it effectively removes two ions from water ($\Delta H = 107\ kcal/mol$)[38,39]. Overall, the reduction of a proton by loss of electrons from $OH^-$ is the simplest redox chemistry that can occur in a sprayed microdroplet and therefore is the primary focus of this work.

Here, we investigate whether the electron transfer process in Eq. (3) becomes more favorable in charged microdroplets, thus providing an explanation for the experimentally observed redox chemistry such as the production of $H_2O_2$. Using simulated nanodroplets, we indeed find that, at between ~20% and ~50% of the Rayleigh limit[40] (i.e. the

maximum charge a droplet can stably accommodate), the hydration enthalpies of $OH^-$ and $H^+$ have decreased to the point that the reaction becomes thermodynamically favorable. We then compute the VIE of $OH^-$ as a function of charge to show that the electron transfer process becomes more favorable with increasing charge. Analogously, we compute the vertical electron affinity (VEA) of $H^+$ and show that it is greatly enhanced in positively charged droplets compared to the neutral case. Finally, while this work utilizes nanodroplets, we present well-defined scaling arguments to show that these results hold and are quantifiable at the microdroplet length scale.

This work emphasizes that the reaction thermodynamics in charged droplets, especially of redox reactions, are not the same as those occurring in bulk solution. We suggest that the mechanism proposed here explains both $H_2O_2$ production and the accelerated organic redox chemistry that occurs in charged microdroplets generated with sonication, electrospray, and gas nebulization experiments, including a water anion that has been observed recently[10]. In addition, it is in principle possible to measure the VIE of $OH^-$ in a charged droplet, and we hope these calculations stimulate future experiments to test our predictions.

## Results
### Thermodynamics of electron transfer in charged droplets

For redox reactions to occur in water, the oxidation reactions typically require the presence of OH[·] while the reduction reactions clearly depend on the ease of electron transfer, both of which are limited by an unfavorable free energy and possible competing reactions[41], but would motivate why experiments observe simultaneous reduction and oxidation of organic species. We have developed a quantum mechanical-statistical mechanical cluster-continuum-charge embedding model that provides estimates of the entropy and enthalpy components of the hydration-free energy of hydronium and hydroxide ions in charged droplets.

It has been shown that the hydration entropy component is determined almost entirely by the cavity produced by an ion or radical[42,43] as measured by its solvent accessible surface area (SASA)[44]. We perform molecular dynamics (MD) simulations of neutral and charged droplets of 4 nm radius containing $H^+$ and $OH^-$ using a reactive force field, ReaxFF/C-Gem[45,46]. ReaxFF/C-Gem has been extensively validated against experiment for pure water properties[45] and surface tension (see Methods), proton hopping mechanisms in bulk water and in reverse micelles[47,48], and relevant here the correct partitioning of $H^+$ and $OH^-$ to surface and bulk regions, respectively[49–51]. From the reactive force field simulations we harvest large ion-containing water clusters, $X(H_2O)_n$, where $n = 35$ and $X=H^+$ or $OH^-$, which is much larger compared to many cluster calculations that typically use between 5-15 explicit solvent molecules[52–54]. Using the harvested ion clusters from the simulations with net droplet charge, Supplementary Note 2 shows that there is not a statistically discernible change in SASA for $OH^-$ while for $H^+$ the SASA increases by only 4%. This is consistent with the observation that hydration entropy is similar for ions and neutral radicals[55]. Hence the hydration entropy is unaffected by excess charges in the environment and the thermodynamic free energy is therefore dominated by the hydration enthalpy.

We compute hydration enthalpies using the thermodynamic cycle developed by Bryantsev and co-workers, which has been successfully applied to the calculation of ion solvation free energies[56].

$$\Delta H_{hyd.}^{(n)} = \Delta H_{gas}^{(n)}[X] + \Delta\Delta H_{solv.}^{(n)}[X] \tag{4}$$

where $X=OH^-$ or $X=H^+$ and the hydration enthalpy of a solvated species $\Delta H_{hyd.}^{(n)}[X]$ is expressed as a sum of two quantities. The first, $\Delta H_{gas}^{(n)}[X]$, is the change in enthalpy of $X(H_2O)_{35}$ compared to the enthalpy of just $(H_2O)_{35}$ in the gas-phase. The second term, $\Delta\Delta H_{solv.}^{(n)}[X]$, is the difference in solvation energy of an $X(H_2O)_{35}$ and $(H_2O)_{35}$ cluster in water.

**Table 1 | Calculated hydration enthalpies of OH⁻ in droplets of increasing total charge with a 4-nm radius**

| Num OH⁻ | $\Delta H_{gas}^{(n)}[OH^-]$ | $\Delta\Delta H_{solv.}^{(n)}[OH^-]$ | $\Delta H_{hyd}^{(n)}[OH^-]$ Expt. -130[59]/ -126[60]/ -117[61] |
|---|---|---|---|
| 1 | −100.9 | −24.2 | −125.7 ± 2.3 |
| 4 | −98.7 | 7.3 | −91.2 ± 2.9 |
| 6 | −99.1 | 26.1 | −73.6 ± 2.9 |
| 8 | −100.6 | 39.2 | −62.0 ± 3.0 |
| 12 | −96.0 | 73.6 | −23.0 ± 3.4 |
| 16 | −96.6 | 106.9 | 9.7 ± 4.1 |

Note that the $\Delta H_{hyd}^{(n)}[OH^-]$ column contains a PV term of −0.59 kcal/mol arising from the difference in volume of an ideal gas and the final density of the liquid. $\Delta H_{gas}^{(n)}[OH^-]$ contains a correction for basis set superposition error (BSSE). Uncertainties are bootstrapped standard errors in the mean of each hydration enthalpy. We report three representative experimental references which are discussed further in the main text.

**Table 2 | Calculated hydration enthalpies of H⁺ in droplets of increasing total charge with a 4-nm radius**

| Num H⁺ Expt. | $\Delta H_{gas}^{(n)}[H^+]$ | $\Delta\Delta H_{solv.}^{(n)}[H^+]$ | $\Delta H_{hyd}^{(n)}[H^+]$ Expt. -258[59]/ -263[60]/ -272[61] |
|---|---|---|---|
| 1 | −234.4 | −28.6 | −263.6 ± 2.4 |
| 4 | −231.1 | 0.7 | −231.0 ± 2.7 |
| 6[a] | −231.9 | 13.9 | −218.0 ± 2.6 |
| 8 | −232.7 | 27.1 | −206.2 ± 2.4 |
| 12 | −239.1 | 59.8 | −179.9 ± 2.4 |
| 16 | −238.4 | 92.6 | −146.4 ± 2.4 |

Note that the $\Delta H_{hyd}^{(n)}[H^+]$ column contains a PV term of −0.59 kcal/mol arising from the difference in volume of an ideal gas and the final density of the liquid. $\Delta H_{gas}^{(n)}[H^+]$ contains a correction for basis set superposition error (BSSE). Uncertainties are bootstrapped standard errors in the mean of each hydration enthalpy. We report three representative experimental references which are discussed further in the main text.
[a]Reported numbers for 6 H⁺ are linear interpolations of the 4 H⁺ and 8 H⁺ entries.

The solvation energy is the change in energy when moving a cluster from the gas-phase to the condensed phase. Note that the solvation enthalpy will be different when solvating an ion in a neutral droplet than when solvating it in a droplet with additional OH⁻ or H⁺ ions, which is what we quantify here. Hence we evaluate the hydration enthalpy of both H⁺ and OH⁻ in neutral and charged nanodroplets to determine if there is a tipping point when the electron transfer process in Eq. (3) goes from unfavorable to favorable. These quantities are calculated from the X(H₂O)₃₅ clusters that are now treated quantum mechanically (QM) with the ωB97M-V/aug-cc-pVDZ level of theory[57,58] (including corrections for basis set superposition error). The QM cluster is then electrostatically embedded in a continuum model to account for any remaining polarization, as well as including Gaussian charges of the ReaxFF/C-Gem model to account for long-ranged permanent electrostatics. More details are provided in Methods.

Tables 1 and 2 provide our calculated hydration enthalpies of the OH⁻ and H⁺ species as a function of excess ions. The first validation of our theoretical model is that $\Delta H_{hyd}[OH^-] + \Delta H_{hyd}[H^+]$ sums to 389.3 ± 3.3 kcal/mol for single ions, in excellent agreement with experimental values of 389 ± 1 kcal/mol[39]. However, there is no experimental consensus on the single-ion hydration enthalpies of H⁺ and OH⁻ since they are not directly measurable quantities. Tables 1 and 2 instead provides three commonly quoted values from Schmid[59], Marcus[60], and Tissandier[61]. The values from Tissandier are determined from extrapolation of ion-cluster data while those of Marcus are based on electrochemical measurements used to infer the absolute hydration enthalpy of H⁺. The values of Schmid rely on the

assumption that the standard hydration entropy of H⁺ and OH⁻ are equal. Ultimately, some kind of assumption must be made to produce a single-ion hydration enthalpy, and these reasonable but different assumptions result in a wide range of values.

Previous calculations of single-ion hydration enthalpies or free energies were typically performed on cold ion-water clusters that included vibrational corrections only at the harmonic level[36,39,56,62] or MD sampling of configurations of water around H⁺ and OH⁻ using ab initio methods where convergence is an issue[63,64]. However, it is known that many structural descriptors of water clusters, (H₂O)ₙ, only approach convergence by around $n = 30$[65], which is much larger than the typical cluster sizes used in the aforementioned cluster-continuum calculations. Perhaps unsurprisingly, those papers tend to report single-ion solvation energies in better agreement with the experimental values of Tissandier[61], which are derived from small ion-water cluster data. Our use of a reactive force field naturally includes anharmonic vibrational contributions to the enthalpy, combined with electronic structure calculations on large clusters (H₂O)₃₅, thereby overcoming the finite size effects from which ab initio MD suffers. For these reasons, we consider that our calculations provide evidence in favor of the hydration enthalpies reported by Marcus[60], which are derived from solution data based on electrochemical measurements.

Having demonstrated the accuracy of our approach based on single ion data, we now are in a position to evaluate the hydration enthalpy changes when the nanodroplet environment becomes increasingly charged. In our MD simulations, we keep the net charge below the Rayleigh limit, $q_{max}$, which describes the maximum charge that can be accommodated by a spherical droplet before Coulomb repulsion overcomes surface tension and breaks the droplet apart:

$$q_{max} = \sqrt{64\pi^2\epsilon_0\gamma R^3}. \qquad (5)$$

The Rayleigh limit depends on the droplet radius, $R$, and the surface tension, $\gamma$; $\epsilon_0$ is the permittivity of free space. For the 4nm radius droplets studied here, the Rayleigh limit occurs at $q_{max} \approx 32$ based on the surface tension of 72 mN/m for water. Hence the thermodynamic analysis and scaling arguments will be analyzed in terms of droplet charge states that are viable before reaching the Rayleigh limit. As also seen in Tables 1 and 2, the hydration enthalpy for both OH⁻ and H⁺ shifts dramatically as the charge increases. But the most important point is to determine when the decrease in hydration enthalpies of both OH⁻ and H⁺ are large enough to overcome the 107 kcal/mol barrier at which Eq. (3) becomes spontaneous. This depends on whether the reaction occurs within a single droplet or across two droplets, as illustrated in Fig. 1, as different thermodynamic pathways are possible depending on the fission process.

Figure 1a, depicts separate droplets with an excess of either H⁺ or OH⁻ and a small remaining concentration of the appropriate counterion such that the reaction could happen within a single droplet. According to Tables 1 and 2, just over 12 OH⁻ ions (~40% of the Rayleigh limit) or just below 16 H⁺ ions (~50% of the Rayleigh limit) is sufficient to overcome the 107 kcal/mol barrier described in Eq. (3). Two issues arise in regards the single droplet with a mixed charge mechanism. First, is that the 40% to 50% of the Rayleigh limit is likely an underestimate of the true crossover since H⁺ or OH⁻ is more stable in an oppositely charged droplet, and thus the barrier in Eq. (3) may not be overcome until greater charge imbalance within a single droplet is reached. Second is a competing pathway whereby OH⁻ and H⁺ would simply recombine to form H₂O versus electron transfer to produce OH· and H·. In general, the electron transfer is expected to be a faster process than ion diffusion such that the single-droplet mechanism depicted in Fig. 1a should be possible.

A second mechanism involves electron transfer between two droplets, each with excess charges of opposite sign. This situation

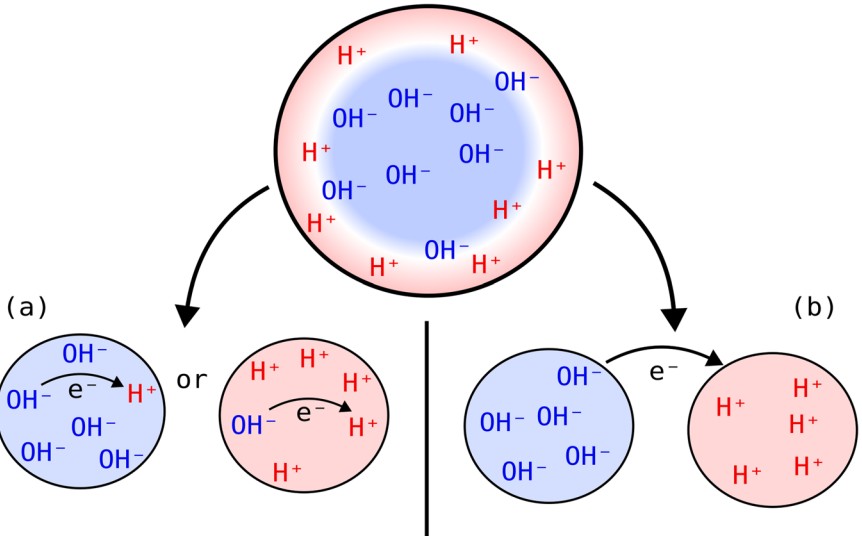

**Fig. 1 | Schematic illustration of two mechanisms for redox chemistry in charged microdroplets.** The ultrasonication, electrospray, as well as gas neb-ulization process forms microdroplets via fission processes that create micro-droplets with a net charge such that redox reactions have different thermodynamic pathways that become viable. **a** According to Tables 1 and 2, within a net-charged droplet containing residual quantities of the counter-ion, an excess of 12 OH- ions (~40% of the Rayleigh limit) or 16 H+ ions (~50% of the Rayleigh limit) are sufficient to overcome the thermodynamic barrier described in Eq. 3. **b** According to Tables 1 and 2, two charged droplets that are near enough so that electrons are transferred from the negative to positive droplet requires an excess of only 8 OH· ions and 8 H+ ions (~20–25% of the Rayleigh limit) to overcome the ther-modynamic barrier.

could arise just after a droplet breaks up or if two droplets of opposite charge collide. According to Tables 1 and 2, when there are 8 OH$^-$ ions in our simulated droplets the shift in $\Delta H_{hyd}^{(n)}[OH^-]$ is 63.7 kcal/mol, while for 8 H$^+$ ions the shift in $\Delta H_{hyd}^{(n)}[H^+]$ is 57.4 kcal/mol, resulting in a total shift of 121.1 kcal/mol that is more than sufficient to overcome the unfavorable thermodynamics of Eq. (3). In fact the cross-over point to overcome the 107 kcal/mol barrier to electron transfer is sufficient with 6 OH$^-$ and 8 H$^+$, in which the total shift in hydration free energy is 109.6 kcal/mol. A droplet with 6–8 charges corresponds to ~20–25% of the Rayleigh limit, which is certainly achievable in experimental lab conditions. The two droplet mechanism provides another reasonable alternative, or could be happening simultaneously with the single droplet case. In either case the electron transfer process is mechan-istically possible in aqueous microdroplets.

## Stabilization of OH• and H• radicals

For microdroplet charge states below the Rayleigh limit, we evaluate the VIE of OH$^-$ and VEA of H$^+$ as both are useful measures of how droplets gain stability by decreasing their charge. The VIEs calculated using our theory as a function of charge are shown in Fig. 2a, and illustrate the important point that electrons are much more weakly bound to OH$^-$ in a charged environment. Fig. 2b shows that electrons are strongly attracted to H$^+$ in a positively charged droplet since the addition of an electron decreases the total droplet charge. Hence while a single excess OH$^-$ or H$^+$ has a very small effect on the computed VIE or VEA, the effect of further excess charges is quite dramatic in lowering the VIE and making the VEA more favorable. Figure 2c shows that the average shifts in the VIE and VEA follow an unscreened Coulomb repulsion starting with 4 OH$^-$ or 4 H$^+$ ions in which the VEA trend is fit to Coulomb's law with $\epsilon = 1$ while the fit to the shift in VIE trend yields $\epsilon = 1.3$. The small differences in apparent dielectric constant mostly reflect the differences in ion distributions of H$^+$ and OH$^-$ in the droplet. The shift in VEA is well-modeled by a completely unscreened Coulomb repulsion since H$^+$ ions are primarily at the surface where the dielectric constant is known to rapidly approach the vacuum value of $\epsilon = 1$[66]. The shift in the VIE is likely better fit by $\epsilon = 1.3$ since OH$^-$ presides both near the surface and in the bulk region where some screening is operative.

The dielectric constant that emerges of 1.0 and 1.3 take into account not only the screening on the electronic process (embodied in the optical dielectric constant of 1.77), but also implicitly includes the molecular response of dipole reorientation around the OH$^-$ and H$^+$ ions (which dominates the static dielectric constant of ~78). This is because the embedded charges from the MD used in the ab initio calculations come from water arrangements that reflect the dipole orientation response around the ions. This is consistent with the fact that for the 2 OH- calculations the VIE is shifted less than one would expect if we referenced the optical dielectric constant of 1, since the embedded charges did provide screening that is more consistent with a dielectric constant of 80.

But starting with 4 OH$^-$ or 4 H$^+$ ions, the shifts in the VIE and VEA with increasing droplet charge imply that there is a rapid onset of dielectric saturation, in which the static dielectric constant of water decreases because water molecules in the presence of ions cannot rearrange their dipoles to screen out large numbers of ionic charges[67,68]. While our calculations do not describe the water response to the neutral radical, this can be neglected as the water response to multiple ions is the dominant effect, and is accounted for in our theory.

For bulk ionic solutions, the Kirkwood-Booth equation[69,70] which models the variation in dielectric constant of water under external fields, predicts that an applied field of 5 MV/cm decreases the dielectric constant of water by ~50%. Simulations have reported that in the immediate vicinity of a single ion in water, the dielectric constant drops to 1 since the first solvation shell is immobilized by strong ion-water interactions, and decays back to the homogeneous bulk value within 1 nm[71]. Depending on the specific ions being considered, the dielectric constant can decrease by half in 5M salt solutions[68]. Addi-tionally, nano-confined water has been shown to develop an asym-metric dielectric profile where the dielectric constant perpendicular to a surface is decreased significantly[72,73]. This effect and more modern literature attribute dielectric saturation as arising from the loss of dipolar correlations in water. For example, in the presence of salt the water hydrogen-bond network is disrupted by the presence of hydra-tion shells of the ions[74], which in turn suppresses the collective dielectric response. This situation is similar to our net charged

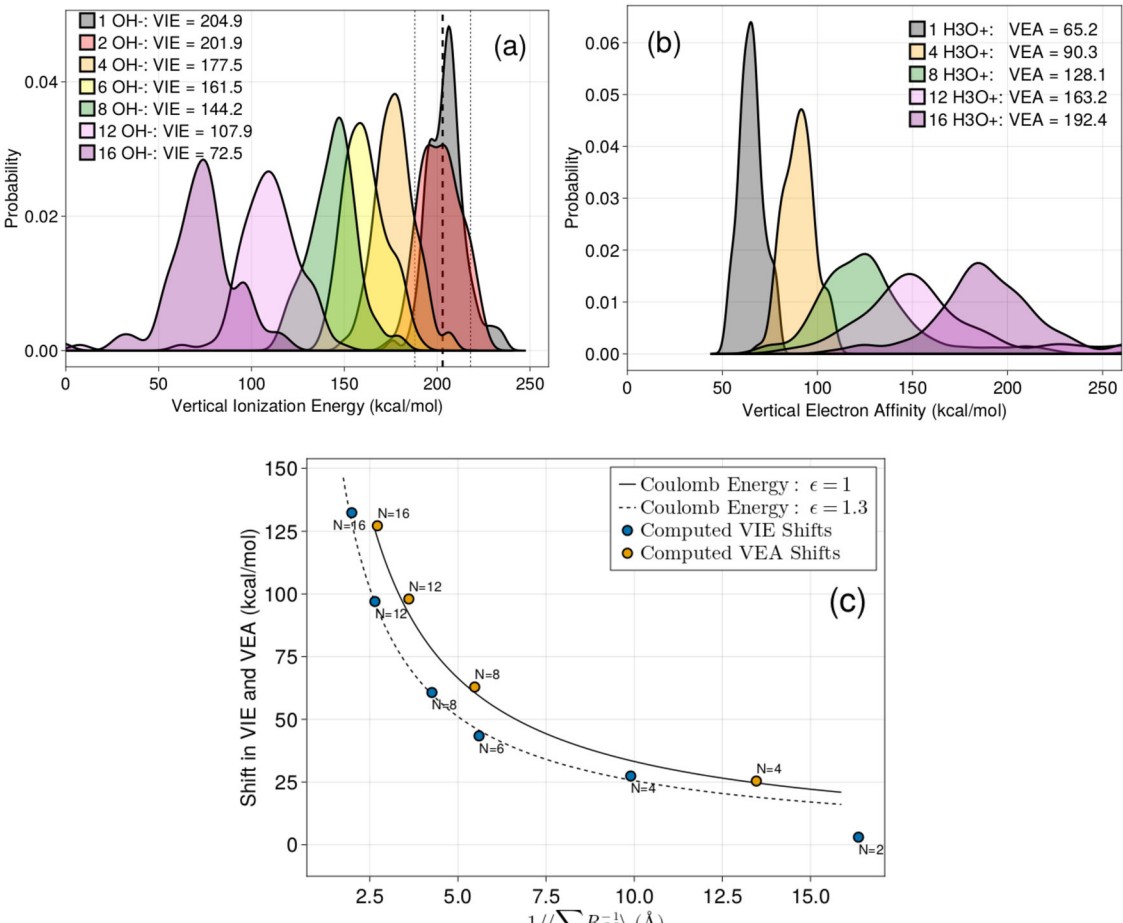

**Fig. 2 | VIE of OH⁻ ions and VEA of H₃O⁺ ions and their relation to an unscreened Coulomb potential as a function of nanodroplet charge.** **a** Computed vertical ionization energies (VIEs) of OH⁻ for 100 configurations of OH⁻ surrounded by 6Å of explicit solvent. The rest of the environment is represented with embedded charges taken from ReaxFF/C-Gem simulations and hence includes both dielectric screening and the effect of excess charges. The legend shows the average VIE for each droplet. The thick vertical dashed line shows the experimental VIE for 0.1M NaOH and the thin dashed lines show the measured full-width at half maximum[94]. A gaussian kernel is applied to each distribution for smoothing. **b** Same as **a** except

for the vertical electron affinity (VEA). Note that, by convention, the VEA is a positive number when energy is released upon addition of an electron. No experimental measurements of this quantitiy are available. **c** The y-axis shows the shift in the VIE and VEA. The curves are Coulomb's law for the repulsion of singly-charged ions in atomic units, $E = 1/\epsilon R$. The value of $\epsilon$ is fit to the data. The x-axis, $1/\langle \sum R_{OO}^{-1} \rangle$, is the inverse of the average sum of inverse distances between each ion taken from our simulations. This quantity computes the effective ion-ion distance as if there were only one additional ion in the system.

droplets in which the overall dipole fluctuations are dominated by the rearrangement of net charge in the droplet, and further exacerbated by the presence of an interface that also breaks up the dipole correlations observed in the bulk solvent. Therefore, there is precedent for large decrements in the apparent dielectric constant in the presence of ions for aqueous systems. However in the case of charged droplets that arise from electrospray or sonication, there is a net imbalance of charge such that dielectric saturation occurs much more readily than in ionic solutions.

Finally, note that the charge-transfer-to-solvent (CTTS) transition for OH⁻ occurs at 6.63 eV or 153 kcal/mol[75]. This means that every VIE in Fig. 2 below a value of about 50 kcal/mol can barrierlessly undergo a CTTS transition. This would indicate that around 50-60% of the Rayleigh limit the equilibrium constant for Eq. (1) is near 1. We suspect this is true because the valence band of OH⁻ will be dramatically affected by surrounding charges while the conduction band of water should be affected much less by the excess charges. This illustrates the drastically different behavior of charged droplets from neutral systems and why one should expect unusual chemistry, especially redox chemistry, to be prevalent in electrosprayed or sonicated droplets.

**Extension from charged nanodroplets to microdroplets**

It is important to address the relevance of our theory to experiments where the droplets are on the scale of micrometers instead of nanometers. When increasing the size of a charged droplet, there are two factors to consider. First, the maximum allowed charge concentration decreases as droplet size increases, in accordance with the Rayleigh limit. Second, while the charge concentrations in nanodroplets exceed the Rayleigh limit for micron-sized droplets, the Coloumbic energy per charge also increases with system size. As we have shown that the shifts in VIE, VEA, and hydration enthalpy are controlled by Coulombic interactions, it stands to reason that similar shifts in these quantities will occur in larger droplets if the Coulombic interaction energy is the same.

To examine this point, we first analyze how the Coulombic interaction energy changes with the charge density $\rho_c$ and droplet radius $R$. The average Coloumbic interaction energy $E_c$ of a charge will be:

$$\frac{\langle E_c \rangle}{N_c} = 4\pi\rho_c \int_0^R r^2 g(r)u(r)dr, \tag{6}$$

where $g(r)$ is the radial distribution function and $N_c$ is the number of charges. Quantitatively evaluating this equation requires knowledge of $g(r)$, however, since we are mainly interested in the scaling behavior for large droplets, we take $g(r) = 1$, which is always true at long range. Because $u(r) = 1/r$ for charges interacting via Coulomb's law (in atomic units), we can write:

$$\frac{\langle E_c \rangle}{N_c} \propto \rho_c R^2 \tag{7}$$

Thus the Coulomb energy increases quadratically with the radius of a droplet. This scaling is well known[76], and results from the long-range nature of Coulombic interactions.

Next, we examine the change in the Rayleigh limit. By dividing the Rayleigh limit (Eq. (5)) by droplet volume, we find that the maximum $\rho_c$ for a droplet of radius $R$ scales as

$$\rho_{max,c} \propto R^{-3/2}. \tag{8}$$

Substituting this charge density into Eq. (7), we find that:

$$\frac{\langle E_c(\rho_c = \rho_{max,c}) \rangle}{N_c} \propto R^{1/2} \tag{9}$$

Thus the Coulombic energy per charge at or below the Rayleigh limit increases with droplet size, which implies that there is greater VIE lowering and increased favorable VEA even at the micron scale. This in turn indicates that the electron transfer reaction in Eq. (3) is favorable in microdroplets at charge concentrations well below the Rayleigh limit.

This scaling argument is consistent with the repeated observation that charged droplets near the Rayleigh limit break down by emitting between 1-5% of their mass but around 20-50% of their charge[77-79]. I.e. a large droplet at the Rayleigh limit emits many very small droplets which have a higher charge density than the original droplet. The scaling arguments made here also explain why smaller droplets are able to accommodate a higher charge density while remaining below the Rayleigh limit. This analysis requires us to assume that ion motions are uncorrelated, which is true at long range, such that the proposed mechanism for $H_2O_2$ production is definitely plausible.

## Discussion

Experiments have observed that many redox reactions are accelerated in aqueous microdroplets, as well as many organic reactions that are found to occur in droplets but do not occur in the bulk liquid. In this work we consider a class of reactions in which the observed oxidation typically occurs via OH· and the observed reductions clearly depend on electron transfer, which we have shown becomes thermodynamically favorable in charged droplets well below the Rayleigh limit that is relevant to hydrogen peroxide formation for example. Based on the success of our calculations in reproducing two well-known experimental quantities, the hydration enthalpy and VIE of single ions, we predict that water droplets with excess charge at ~ 20–50% of the Rayleigh limit makes Eq. (3) spontaneous. We interpret our large shifts in VIE/VEA from high-quality electronic structure calculations as evidence that in the presence of excess charges, water is unable to screen out the field of other like charges, resulting in an apparent dielectric constant of nearly 1 that is unlike that of bulk ionic solutions that have counter charges. Hence the VIE/VEA shifts we measure ought to stimulate generalizations of existing dielectric theories of ionic solutions to the case of charged liquids.

The fact that the electron transfer from OH$^-$ to H$^+$ is favorable for the crucial step of radical formation supports the production of $H_2O_2$ that has been observed in microdroplets prepared with ultrasonication and gas nebulization[13,14], albeit at lower concentrations than previously reported[80,81]. Furthermore, the conclusions drawn from nanodroplet calculations are fully extensible to the tens of microns lengthscale of real experiments. In particular, in the long-range limit, one should expect the same Coulomb repulsion per ion at a smaller fraction of the Rayleigh limit. This means that the electron transfer process measured by the VIE and VEA will become thermodynamically favorable in large and charged droplets before the Rayleigh limit is reached. In conclusion, our work provides a thermodynamic explanation for why many organic redox reactions are accelerated in microdroplets, especially those that result in the addition of hydroxyl radicals or formation of anion radicals[6,10,11]. Our work is also relevant to the recent observation of reduction of inorganic cations in sprayed water droplets[7,12].

## Methods

For the cluster-continuum model, we generate ion-water clusters by running molecular dynamics simulations on droplets of water with a 4 nm radius using the ReaxFF/CGeM reactive force field which has proven successful in modeling the energetics and dynamics of H$^+$ and OH$^-$[45,46]. From these simulations we harvest 100 configurations of $X(H_2O)_{35}$ from a simulation of a neutral droplet, as well as charged droplets prepared with 1, 2, 4, 6, 8, 12, and 16 OH$^-$ ions as well as 1, 4, 8, 12, and 16 H$^+$ ions. PACKMOL is used to generate initial conditions for all droplets[82]. All simulations are equilibrated at 293K for 1 ns, then an additional 250 ps of simulation time is used to harvest the configurations. All configurations are chosen at random without replacement. Simulations of the same conditions, but with no solute present (i.e., a pure water droplet), are used to extract 200 configurations of $(H_2O)_{35}$.

To further validate the ReaxFF/CGem model, we have computed its surface tension using 512 and 1024 water molecules in a slab geometry with the computational procedure reported by Muniz and co-workers[83]. We find the computed surface tensions to be 60 mN/m without long-ranged corrections, and 64 mN/m with an analytical correction proposed by Vega[84], in reasonable agreement with the 72 mN/m from experiment and for fixed charge as well as polarizable models of water[84-86]. Further details are provided in Supplementary Note 1.

With all of these configurations as a function of droplet charge in hand, we compute the energy of the clusters taken from simulation at the $\omega$B97M-V/aug-cc-pVDZ level of theory using Q-Chem[57,58,87]. Then, the binding energy contribution to the enthalpy can be written as,

$$\Delta H_{gas}^{(n)}[X] = \langle E[X(H_2O)_n] \rangle - \langle E[(H_2O)_n] \rangle - E[X] \tag{10}$$

where $\langle E[X(H_2O)n] \rangle$ is the average energy of the 100 $X(H_2O)_{35}$ clusters taken from our MD simulations and $\langle E[(H_2O)_n] \rangle$ is the same for the 200 $(H_2O)_{35}$ clusters. $E[X]$ is the gas-phase optimized energy of the ion X. Notice that $\Delta H^{(n)}[X]_g$ is the average difference in binding energy of the unrelaxed ion-water clusters and water clusters. This means we naturally capture the vibrational contributions to the enthalpy including anharmonicity. This is one advantage of our approach over cluster-continuum models which rely on locating the global minimum structure. Our approach also naturally includes the cavitation energy.

The solvation contribution is computed using a combination of polarizable continuum calculations with the COSMO model[88] and charge embedding. COSMO enables us to capture the cluster-continuum polarization, but is incapable of describing the excess charges in the environment. In order to capture the contribution of excess charges, we compute the solvation energy using embedded charges taken from the CGeM model which are represented as point charges in the electronic structure calculation. The shift in solvation energy is then described as the difference in point-charge embedded solvation energies for the single-ion case and the multiple-ion case.

$$\Delta H_{solv.}^{(n)}[X] = \langle E[X(H_2O)_n] \rangle_{cosmo} + (\langle E[X(H_2O)_n] \rangle_k - \langle E[X(H_2O)_n] \rangle_0)$$

$$\tag{11}$$

where $k$ is the net charge. We can then compute the final solvation contribution to the hydration enthalpy as,

$$\Delta\Delta H_{solv.}^{(n)}[X] = \Delta H_{solv.}^{(n)}[X] - \langle E[(H_2O)_n]\rangle_{cosmo} \qquad (12)$$

The total enthalpy of hydration described by Eq. (4) reported in the main text is then the sum of the results of Eqs. (10) and (12). Note that we do not include a much discussed state correction[56,89] which has given rise to confusion, because it only arises in the entropy term.

We also compute a correction for basis set superposition error (BSSE) which is included in our hydration enthalpies. We do this using a pairwise approximation of the usual counterpoise correction[90], as validated in past work for both water clusters[91] and ion-water clusters[92]. The reason we use a pairwise approximation to BSSE is that computing a full counterpoise correction for a cluster of 35 molecules requires a full system calculation and 35 calculations with the entire set of basis functions. This is computationally demanding and instead an estimated correction of just a couple of kcal/mol will be sufficiently accurate without need for the brute force calculation. Full details of our approach to correcting for BSSE can be found in Supplementary Note 3.

Finally, to gain insight into the electron transfer process which might occur in these reactions, we compute the vertical ionization energy (VIE) of OH⁻ in these same charged droplets.

$$E_{VIE}[OH^-] = E[OH^-(H_2O)_n] - E[OH^\bullet(H_2O)_n] \qquad (13)$$

When computing the VIE, we make two minor modifications to the protocol for computing hydration enthalpies. First, we use an aug-cc-pVTZ basis set on OH⁻ and OH˙ while using aug-cc-pVDZ for the surrounding solvent. Second, rather than using a fixed number of waters, we use a fixed radius of 6 Å to sample the explicit solvent neighboring OH⁻. We use charge embedding with charges taken from the MD simulations to describe the environment surrounding the ion-water cluster. We do not use the same solvation approach as for the hydration enthalpy case since any errors at the edge of the cluster will cancel almost perfectly since the VIE is a difference in energy of the same cluster with different charge. Additionally, we always compute the gas-phase VIE and use the orbitals from each of these calculations as the initial guess orbitals for the calculation with explicit charges. This ensures that ionization occurs from OH⁻, which we can validate by looking at the mulliken spin density on OH˙. This overall approach to computing VIEs, including the choice of functional and basis set, has been applied successfully to many other ions[93].

We also compute the vertical electron affinity (VEA) which is given by,

$$E_{VEA}[H_3O^+] = E[H_3O^+(H_2O)_n] - E[H_3O^\bullet(H_2O)_n] \qquad (14)$$

We follow the same protocol as just described for the VIE when computing the VEA. The only difference being that we place aug-cc-pVTZ basis functions on the atoms of the central $H_3O^+$ ion. Notice that the way we have written Eq. (14) follows the sign convention that a positive number means the radical state is lower in energy than the cationic state.

It should be noted that we compute vertical quantities, as opposed to adiabatic ones, which means the VIE and VEA calculations do not account for solvent relaxation upon detachment and attachment of electrons, respectively. This removes the need to model the dynamics of solvated radicals. Additionally, VIEs are experimentally measurable so we can use the VIE calculations to confirm the reliability of our methodology (*a posteriori*). Note that our hydration enthalpy calculations do, of course, account for solvent response to the presence of an ion.

## Reporting summary
Further information on research design is available in the Nature Portfolio Reporting Summary linked to this article.

## Data availability
The coordinates of atoms and embedding charges used in all electronic structure calculations are available at https://figshare.com/projects/Charged_Microdroplets/193307. Source data for Tables 1 and 2 and Fig. 2 are available with this manuscript. Source data are provided with this paper.

## Code availability
All analysis scripts are in a private github repo but are available upon request.

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

## Acknowledgements

We thank the Air Force Office of Scientific Research through the Multidisciplinary University Research Initiative (MURI) program under AFOSR Award No. FA9550-21-1-0170 for the microdroplet chemistry application. The reactive force field work was supported by the National Science Foundation under Grant CHE-2313791. This work used computational resources provided by the National Energy Research Scientific Computing Center (NERSC), a U.S. Department of Energy Office of Science User Facility operated under Contract No. DE-AC02-05CH11231.

## Author contributions

J.P.H. and T.H-G. conceived the theme, J.P.H. formulated the level of theory and performed all ab initio calculations. J.P.H. and R.A.L. developed the nano- to microscale scaling arguments. J.P.H. and T.H-G. wrote the manuscript, and J.P.H., R.A.L., and T.H-G. contributed to all insights through extensive discussion.

## Competing interests

The authors declare no competing interests.
