## [Peer Review File · Nature Communications]

The Role of Charge in Microdroplet Redox ChemistryReviewer #1 (Remarks to the Author):

The authors present compelling arguments that charged droplets promote redox chemistry, a topic of great personal interest to this reviewer. The production of radicals, OH and H, then leads to a host of different chemical reactions offering explanations to two important questions: (1) why water microdroplets are highly reactive whereas bulk water is not; and (2) why water microdroplets can produce products not possible in bulk water. This manuscript is judged by this reviewer to be of major importance should be published without much delay.

However, there are some additional features that I believe that should be included to help readers and to make a yet stronger case. One is the experimental observation of H and H₂. See X. Chen, Y. Xia, Z. Zhang, L. Hua, X. Jia, F. Wang, and R. N. Zare, "Hydrocarbon degradation by contact with anoxic water microdroplets," *J. Am. Chem. Soc.* 145, 21538–21545 (2023). Another is that the electron transfer reaction given in Equation (3) is exoergic at the interface. See Colussi, A. J. Mechanism of hydrogen peroxide formation on sprayed water microdroplets, *J. Am. Chem. Soc.*, 2023, 145, 16315-16317. Finally, I think it is worth mentioning that the electric field strength at the interface has been experimentally measured to be ten to the ninth volts per meter, and this is a lower bound as the experimental measurement averaged over the interface and some of the droplet interior; see H. Xiong, J. K. Lee, R. N. Zare, and W. Min, "Strong Electric Field Observed at the Interface of Aqueous Microdroplets," *J. Phys. Chem. Lett.* 11, 7423–7428 (2020).

I congratulate the authors on this truly outstanding contribution to our understanding of water microdroplet chemistry.

Sincerely yours,

Richard N. Zare

Reviewer #2 (Remarks to the Author):

The manuscript entitled "The Role of Charge in Microdroplet Redox Chemistry" has presented the theoretical work on the noticeable enhanced stabilization of radical OH* in/on the water droplet compared to its thermodynamic stability in bulk water. The theoretical work was based on both the classic molecular dynamics simulation and quantum chemistry computer simulations. The research subject itself is interesting. The conclusion, if it is valid, will be important too. However, there are also major concerns on the theoretical simulation approach and physics analysis on the "dielectric saturation" on the droplet with hydrated charges. Based on the quality and importance of this work, therefore, I would not recommend its publication in *Nature Communications*, instead, it is more suitable for publication in *Scientific Reports* or journals focusing on the field. The comments are summarized as follows:

1. In the computer simulations of the trajectories of the droplets as well as water droplets with solvated ions and charges, the authors essentially adopted the classical force fields. Later, based on the configurations obtained from the classical force field, they used more physical and more sophisticated quantum chemistry method carried out the electronic structure calculations. It is well known that hydroxide (OH⁻) as well as hydronium (H₃O⁺) will diffuse very fast in aqueous environment via the proton transfers. It is also known that the capability of classical force field is very limited because its lack of capture of electronic structure during the proton transfer. On the other hand, the correct description of proton transfer will be crucial in giving an accurate description of the thermodynamic free energy surface of the solvated OH⁻ and H₃O⁺ (please refer to previous work by the group of M. Parrinello). Unfortunately, the current work neglected the modeling and the discussion of the important process of solvated OH⁻ and H₃O⁺ in water droplets. Therefore, the thermodynamic stability might not be accurate. The above effect can be modeled by either the ab initio molecular dynamics or quantum chemistry approaches. It is understood that such calculations were very expensive, but now, with the state-of-the-art machine learning approach, the above calculations can now be done much more efficiently.

2. The section "Stabilization of OH• and H• radicals" is one of the most part in this work which

elucidates the underlying physics. In my opinion, this part could be strengthened. In particular, the discussion connecting the VIE and VEA to the computed "dielectric saturation" theory is not very clear. According to modern theory of polarization and the following development of methods in computing the dielectric constant, the dielectric constant in water and aqueous solutions should be understood as a collective dipole fluctuation on the underlying hydrogen bond network. Kirkwood and Pople have developed the modernized theory for the dielectric constant and essentially, the changes in dielectric constants in solutions should be described by the drop of Kirkwood G factor. In this work, the authors used a universal and empirical dielectric constant (1.0 for OH⁻ and 1.3 for H⁺) for a series of very different concentrations. The above will imply the water droplets with large variations of solvated ions have essentially the same Kirkwood G factor. The above assumptions are not reasonable, and highly unlikely in reality. To further strengthen this part, a more detailed analysis based on Kirkwood theory should be considered.

3. There are a few typos, for example, in third paragraph of section of "Main", the reference should be added after the sentence "As shown by Hao and co-workers". In the section of "Thermodynamics of electron transfer in charged droplets", in the sentence, "When we factor in the effect of net droplet charge on the SASA of the 96 ions, Supplementary Fig. ?? shows that there is not a statistically discernible change in SASA 97 for OH⁻ while for H⁺ the SASA increases by only 4%.", the question mark should be replaced by the actual number in the supporting material.

Reviewer #3 (Remarks to the Author):

This is an interesting manuscript that proposes the spontaneous formation of radical species (*OH and *H) in charged aqueous droplets. Unfortunately, the data are not particularly well presented, and readers are faced with more questions than answers. After revisions, this work might become acceptable for publication, but the authors should strive for a clearer presentation of their ideas, concepts, and findings. Also, the sweeping statements in the Conclusions section, claiming that the data shown here can explain a wide range of droplet-mediated rate enhancements, should be toned down.

Specific Comments:

p. 2: "One fascinating example is the production of hydrogen peroxide (H₂O₂) in sprayed microdroplets" While an interesting observation, this H₂O₂ production is not a direct example of rate enhancement. Also, the second part of this sentence ("... which is well above ...") is not relevant and should be deleted.

p. 2: "the preparation of the droplets leads to charged microdroplets using ultrasonication and electrospray," this is a poorly worded sentence.

p. 2: "that charged droplets occur not only in the laboratory^{14,21 52 53} but in nature as well^{22,23}," references to thunder clouds seem to be missing.

p. 2: "smaller droplets tend to have a more negative charge while larger droplets tend to have a more positive charge" This is not always true. For example, the preceding sentence mentioned electrospray, where the sign (positive vs negative) is governed by the voltage settings of the sprayer, not by the droplet size.

p. 2: "A microdroplet with a net charge is not the same as increasing ionic concentrations in a bulk liquid" poor grammar and language

p. 2: "the equilibrium constant associated with Eq. 1 is negligibly small since it involves the desolvation of OH⁻," please make it more clear what their key point is, *OH is still solvated, as indicated by the "aq" in eq. 1.

p. 2: "if it were produced it would rapidly reduce any available species such as an organic molecule

or even protons produced by water autoionization" It seems that recombination with OH* would be the most likely outcome. Why is this not mentioned?

p. 3: "an explanation for the experimentally observed redox chemistry" remind readers what experiments you are referring to. Presumably H₂O₂ formation?

p. 3: "Supplementary Fig. ??"

Check abstract and main text and use VEA or EA consistently. Or is there a difference? Not clear.

p. 3: "For redox reactions to occur in water the oxidation reactions typically require the presence of OH• while the reduction reactions clearly depend on the ease of electron transfer, which in the bulk phase are limited by an unfavorable free energy." Unclear how readers are to deal with such sweeping opening statements. Under many conditions, the *OH will most likely abstract a hydrogen from somewhere (i.e. there will be no electron transfer). This is even true for pure water [Codorniu-Hernandez, E., Kusalik, P.G.: Mobility Mechanism of Hydroxyl Radicals in Aqueous Solution via Hydrogen Transfer. J. Am. Chem. Soc. 134, 532-538 (2012)]

Table 1 layout is awkward, as Num H⁺ seems to be directly linked to Expt

The Table 1 data of hydration enthalpies are based on 4 nm droplets (but readers only find this information after digging through the Methods). What is the positioning of OH⁻ and H⁺ in these droplets? Can the authors demonstrate that their placement of these ions is appropriate and fully equilibrated? After all, positioning of H⁺ and OH⁻ in aqueous droplets is a non-trivial problem, see work by Greg Voth and others. For droplets containing multiple ions, some may be in the interior and others will be on the surface. Yet, the tiny error bars in Table 1 suggest that all ions share the same enthalpies. Why? Droplet structures and ion positions have to be discussed in much greater detail, before readers can be convinced that the Table 1 energetics are reliable.

p. 5: "the Rayleigh limit occurs at $q_{max} \approx 32$ based on the surface tension of 72 mN/m for water." How well does that water model used here reproduce this experimental surface tension? Many water models perform quite poorly. How does this aspect affect the validity of the data reported here? [Vega, C., de Miguel, E.: Surface tension of the most popular models of water by using the test-area simulation method. J. Chem. Phys. 126, 154707 (2007)]

p. 6: "Fig. 2a, and illustrate the important point that electrons are much more weakly bound to OH⁻ in a charged environment." Lots of open questions here. Any electron transfer will be associated with major changes in the hydration environment, with Eigen vs. Zundel and other hydration motifs. None of this is discussed in this work. I guess the authors try to get around this by using "vertical" energies, but it is not clear in how far this vertical approach yields data that are reliable and relevant.

On a related note, for a journal with a general audience, the concept of "vertical" (and its appropriateness for the calculations of this work) has to be explained much more clearly.

p. 9: "It has been observed that many redox reactions are accelerated in aqueous microdroplets, ... The observed oxidation typically occurs via OH•" AND "In summary, our work helps explain why there are many organic redox reactions that are accelerated in microdroplets^{6,10,11}" This is a massive oversimplification. The authors have to tone down their claims. While *OH may be involved in some types of droplet acceleration, the favorability of *OH formation seen here will only apply to a small sub-set of these reactions. There are many droplet redox processes that do not involve *OH [see for example Scheme 1 in Chen, C.J., Williams, E.R.: The role of analyte concentration in accelerated reaction rates in evaporating droplets. Chem. Sci. 14, 4704-4713 (2023)].

Figure 1 is highly speculative. It seems to suggest that droplet fission is an essential component of the model proposed in this work. Why would process (a) not occur in the larger droplet shown at the top? Instead of showing such speculative cartoons as the first (!) figure in this manuscript, it would be better to start off with actual data, such as droplet structures used for generating the

data in Table 1. Please use additional and better figures to explain the strategy used in this work, as well as illustrating the findings.

Reviewer #1 (Remarks to the Author):

The authors present compelling arguments that charged droplets promote redox chemistry, a topic of great personal interest to this reviewer. The production of radicals, OH and H, then leads to a host of different chemical reactions offering explanations to two important questions: (1) why water microdroplets are highly reactive whereas bulk water is not; and (2) why water microdroplets can produce products not possible in bulk water. This manuscript is judged by this reviewer to be of major importance should be published without much delay.

However, there are some additional features that I believe that should be included to help readers and to make a yet stronger case. One is the experimental observation of H and H₂. See X. Chen, Y. Xia, Z. Zhang, L. Hua, X. Jia, F. Wang, and R. N. Zare, "Hydrocarbon degradation by contact with anoxic water microdroplets," *J. Am. Chem. Soc.* 145, 21538–21545 (2023). Another is that the electron transfer reaction given in Equation (3) is exoergic at the interface. See Colussi, A. J. Mechanism of hydrogen peroxide formation on sprayed water microdroplets, *J. Am. Chem. Soc.*, 2023, 145, 16315-16317. Finally, I think it is worth mentioning that the electric field strength at the interface has been experimentally measured to be ten to the ninth volts per meter, and this is a lower bound as the experimental measurement averaged over the interface and some of the droplet interior; see H. Xiong, J. K. Lee, R. N. Zare, and W. Min, "Strong Electric Field Observed at the Interface of Aqueous Microdroplets," *J. Phys. Chem. Lett.* 11, 7423–7428 (2020).

I congratulate the authors on this truly outstanding contribution to our understanding of water microdroplet chemistry.

We thank the reviewer for their praise of our work. We have added references to each of the papers mentioned in the introduction from lines 41 to 55 along with some additional discussion of the possible differences between oil-water and air-water interfaces which could be relevant to the experimental results referenced by the reviewer.

We have added the following text:

Very recently, however, H₂ gas has been observed to be produced by oil-water emulsions along with observation of H• and OH• radicals using electron paramagnetic resonance.⁽²²⁾ These observations provide important additional evidence that water-oil interfaces can drive redox chemistry. There is some debate about the similarity of oil-water emulsions and the air-water interface^(23–25), but the work of Chen⁽²²⁾ confirms that redox chemistry can certainly occur at oil-water interfaces formed by emulsification.

Reviewer #2 (Remarks to the Author):

The manuscript entitled "The Role of Charge in Microdroplet Redox Chemistry" has presented the theoretical work on the noticeable enhanced stabilization of radical OH* in/on the water droplet compared to its thermodynamic stability in bulk water. The theoretical work was based on both the classic molecular dynamics simulation and quantum chemistry computer simulations. The research subject itself is interesting. The conclusion, if it is valid, will be important too. However, there are also major concerns on the theoretical simulation approach and physics analysis on the "dielectric saturation" on the droplet with hydrated charges. Based on the quality and importance of this work, therefore, I would not recommend its publication in *Nature Communications*, instead, it is more suitable for publication in *Scientific Reports* or journals focusing on the field. The comments are summarized as follows:

1. In the computer simulations of the trajectories of the droplets as well as water droplets with solvated ions and charges, the authors essentially adopted the classical force fields. Later, based on the configurations obtained from the classical force field, they used more physical and more sophisticated quantum chemistry method carried out the electronic structure calculations. It is well known that hydroxide (OH⁻) as well as hydronium (H₃O⁺) will diffuse very fast in aqueous environment via the proton transfers. It is also known that the capability of classical force field is very limited because its lack of capture of electronic structure during the proton transfer. On the other hand, the correct description of proton transfer will be crucial in giving an accurate description of the thermodynamic free energy surface of the solvated OH⁻ and H₃O⁺ (please refer to previous work by the group of M. Parrinello). Unfortunately, the current work neglected the modeling and the discussion of the important process of solvated OH⁻ and H₃O⁺ in water droplets. Therefore, the thermodynamic stability might not be accurate. The above effect can be modeled by either the ab initio molecular dynamics or quantum chemistry approaches. It is understood that such calculations were

very expensive, but now, with the state-of-the-art machine learning approach, the above calculations can now be done much more efficiently.

We agree with the reviewer that using *ab initio* molecular dynamics (AIMD) is not feasible in this situation. The system sizes required (around 8500 molecules) are very large compared to those routinely used in AIMD (around 128 molecules). The reviewer suggests that instead of AIMD, we could construct a machine-learned potential for water, hydronium, and hydroxide. We are experienced in the construction of machine-learned potentials and can say with confidence that this is a much harder problem than implied, especially for this paper. It has to describe both interfacial and fully solvated configurations and many DFT functionals struggle to accurately describe water and much less is known about the accuracy of functionals in modeling the properties of hydronium and hydroxide. Hence data generation with AIMD remains a problem and is not feasible here.

Fortunately, and perhaps we did not make clear enough, we are using a reactive force field for water, ReaxFF/C-GeM that does in fact describe hydronium and hydroxide and proton transfer. The model chosen has been shown to capture the salient features of liquid water accurately, the faster diffusion of hydronium and Grotthus proton hopping mechanism, the hypercoordination of hydroxide, and its slower diffusion. It gets the surface potential correct, reproduces the correct electric field value and preference for hydronium at the air-water interface. Furthermore we showed that the configurations used, when combined with accurate electronic structure, are able to reproduce both hydration enthalpies and VIE distributions of a single hydroxide as per experiment. Therefore, we are confident in the accuracy of the force field as a means of generating configurations to be used for electronic structure calculations which the reviewer agrees are more physical and more sophisticated.

We have moved some of the Methods section to the main body of the paper as follows:

We have developed a quantum mechanical-statistical mechanical cluster-continuum-charge embedding model that provides estimates of the entropy and enthalpy components of the hydration free energy of hydronium and hydroxide ions in charged droplets.

We perform molecular dynamics (MD) simulations of neutral and charged droplets of 4 nm radius containing H^+ and OH^- using a reactive force field, ReaxFF/C-Gem(44,45). ReaxFF/C-Gem has been extensively validated against experiment for pure water properties(44) and surface tension (see Methods), proton hopping mechanisms in bulk water and in reverse micelles,(46,47), and relevant here the correct partitioning of H^+ and OH^- to surface and bulk regions, respectively.(48–50) From the reactive force field simulations we harvest large ion-containing water clusters, $X(H_2O)_n$, where $n = 35$ and $X=H^+$ or OH^- , which is much larger compared to many cluster calculations that typically use between 5-15 explicit solvent molecules.(51–53)

These quantities are calculated from the $X(H_2O)_{35}$ clusters that are now treated quantum mechanically (QM) with the $\omega B97M-V/aug-cc-pVDZ$ level of theory(56,57) (including corrections for basis set superposition error). The QM cluster is then electrostatically embedded in a continuum model to account for any remaining polarization, as well as including Gaussian charges of the ReaxFF/C-Gem model to account for long-ranged permanent electrostatics. More details are provided in Methods

2. The section “Stabilization of OH^\bullet and H^\bullet radicals” is one of the most part in this work which elucidates the underlying physics. In my opinion, this part could be strengthened. In particular, the discussion connecting the VIE and VEA to the computed “dielectric saturation” theory is not very clear. According to modern theory of polarization and the following development of methods in computing the dielectric constant, the dielectric constant in water and aqueous solutions should be understood as a collective dipole fluctuation on the underlying hydrogen bond network. Kirkwood and Pople have developed the modernized theory for the dielectric constant and essentially, the changes in dielectric constants in solutions should be described by the drop of Kirkwood G factor. In this work, the authors used a universal and empirical dielectric

constant (1.0 for OH⁻ and 1.3 for H⁺) for a serial of very different concentrations. The above will imply the water droplets with large variations of solvated ions have essentially the same Kirkwood G factor. The above assumptions are not reasonable, and highly unlikely in reality. To further strengthen this part, a more detailed analysis based on Kirkwood theory should be considered.

We believe that this comment stems from a misunderstanding of our calculations shown in Figure 2c in our paper. These dielectric constants are never used in our calculations. Rather, in Figure 2c we plot the average shift in VIE/VEA from electronic structure against the inverse of a sum over $1/r$, and the dielectric constant is just a fitting parameter that emerges to be ~ 1 . Hence the dielectric constants reported in Figure 2c naturally emerge from the electronic structure calculations carried out and are therefore not used in the methodology.

We have clarified the text as follows.

Fig. 2c shows that the average shifts in the VIE and VEA follow an unscreened Coulomb repulsion starting with 4 OH⁻ or 4 H⁺ ions in which the VEA trend is fit to Coulomb's law with $\epsilon = 1$ while the fit to the shift in VIE trend yields $\epsilon = 1.3$. The small differences in apparent dielectric constant mostly reflect the differences in ion distributions of H⁺ and OH⁻ in the droplet. The shift in VEA is well-modeled by a completely unscreened Coulomb repulsion since H⁺ ions are primarily at the surface where the dielectric constant is known to rapidly approach the vacuum value of $\epsilon = 1$. The shift in the VIE is likely better fit by $\epsilon = 1.3$ since OH⁻ presides both near the surface and in the bulk region where some screening is operative.

However we thank the reviewer for raising this important issue! Why do we get a dielectric constant close to 1? This is because most classical theories of dielectric saturation only deal with neutral ionic solutions (i.e. with counter charges). As can be seen in Figure 2c, when there is only a single excess ion, there is still a lot of dielectric screening due to rearrangement of water dipoles (as it is not on the curve). Beyond that the net charge of a droplet (without a balancing counterion) causes rapid dielectric saturation. Hence our high quality electronic structure calculations should motivate an extension of classical theories of dielectric saturation to the case of charged systems, as we are not aware of any existing dielectric theories that account for charged droplets.

We have revised the text as follows.

The shifts in the VIE and VAE with increasing droplet charge therefore imply that there is a rapid onset of dielectric saturation, in which the dielectric constant of water decreases because water molecules in the presence of ions cannot rearrange their dipoles to screen out ionic charges.^(66,67) We are not aware of any model that describes the variation in dielectric constant for systems with a net charge. For bulk ionic solutions, the Kirkwood-Booth equation^(69,70) which models the variation in dielectric constant of water under external fields, predicts that an applied field of 5 MV/cm decreases the dielectric constant of water by 50%. Simulations have reported that in the immediate vicinity of a single ion in water, the dielectric constant drops to 1 since the first solvation shell is immobilized by strong ion-water interactions, and decays back to the homogeneous bulk value within 5-10Å.⁽⁷¹⁾ Depending on the specific ions being considered, the dielectric constant decreases by half in 5M salt solutions.⁽⁶⁷⁾ Therefore, there is precedent for large decrements in the apparent dielectric constant in the presence of ions for aqueous systems. However in the case of charged droplets that arise from electrospray or sonication, there is a net imbalance of charge such that dielectric saturation occurs much more readily than in ionic solutions.

3. There are a few typos, for example, in third paragraph of section of "Main", the reference should be added after the sentence "As shown by Hao and co-workers". In the section of "Thermodynamics of electron transfer in charged droplets", in the sentence, "When we factor in the effect of net droplet charge on the SASA of the 96 ions, Supplementary Fig. ?? shows that there is not a statistically discernible change in SASA 97 for OH⁻ while for H⁺ the SASA increases by only 4%.", the question mark should be replaced by the actual number in the supporting material.

Thank you. We have corrected these mistakes.

Reviewer #3 (Remarks to the Author):

This is an interesting manuscript that proposes the spontaneous formation of radical species (*OH and *H) in charged aqueous droplets. Unfortunately, the data are not particularly well presented, and readers are faced with more questions than answers. After revisions, this work might become acceptable for publication, but the authors should strive for a clearer presentation of their ideas, concepts, and findings. Also, the sweeping statements in the Conclusions section, claiming that the data shown here can explain a wide range of droplet-mediated rate enhancements, should be toned down.

p. 2: “One fascinating example is the production of hydrogen peroxide (H₂O₂) in sprayed microdroplets” While an interesting observation, this H₂O₂ production is not a direct example of rate enhancement. Also, the second part of this sentence (“... which is well above ...”) is not relevant and should be deleted.

We have updated the text to read:

Of particular interest are the many observations of redox chemistry occurring at accelerated rates(6,8–12) as well as the observation of hydrogen peroxide (H₂O₂) formation in sprayed microdroplets at concentrations of $\approx 1\text{--}2\ \mu\text{M}$.(13,14) Since production of H₂O₂ in water is thermodynamically disfavored, these experiments collectively point to microdroplets having strong redox properties. At present, however, the molecular origins of the strong redox properties of microdroplets are unclear.

p. 2: “the preparation of the droplets leads to charged microdroplets using ultrasonication and electrospray,” this is a poorly worded sentence.

We have re-worded the text to read:

Another important aspect of microdroplet chemistry is that in the preparation of droplets using ultrasonication, electrospray, as well as gas nebulization, microdroplets are formed with a net charge.(14)

p. 2: “that charged droplets occur not only in the laboratory14,21 52 53 but in nature as well22,23,” references to thunder clouds seem to be missing.

We have added an appropriate reference for the case of charging of thunderclouds and subsequent discharge.

p. 2: “smaller droplets tend to have a more negative charge while larger droplets tend to have a more positive charge” This is not always true. For example, the preceding sentence mentioned electrospray, where the sign (positive vs negative) is governed by the voltage settings of the sprayer, not by the droplet size.

We cited papers that support this idea, i.e. that occur by processes where there is not a voltage bias. But we agree with the reviewer that we should avoid confusion, and have modified the sentence to read,

There is very good evidence that charged droplets occur not only in the laboratory(14,26) but in nature as well(27–29), and there are consistent observations that smaller droplets tend to have a more negative charge while larger droplets tend to have a more positive charge when produced via natural processes.(27,28,30,31)

p. 2: “A microdroplet with a net charge is not the same as increasing ionic concentrations in a bulk liquid” poor grammar and language

In addition to grammar, we have expanded the text to emphasize the importance of charge for redox chemistry.

Furthermore, a microdroplet with a net charge is not the same as an ionic solution which always contains counterions. While ionic solutions can modulate the properties and reaction thermodynamics compared to neutral solutions(32–34), the net charge carried by a microdroplet offers the possibility of a more dramatic alteration of the reaction thermodynamics. This motivates us to examine redox reaction activity in a charged microdroplet as an origin of difference relative to a bulk ionic system.

p. 2: “the equilibrium constant associated with Eq. 1 is negligibly small since it involves the desolvation of OH⁻,” please make it more clear what they key point is, *OH is still solvated, as indicated by the “aq” in eq. 1.

The point of the sentence should now be very clear with this modification:

Ordinarily, the equilibrium constant associated with Eq. 1 is negligibly small since OH⁻ is more strongly solvated than the electron, which is why H₂O₂ is not normally produced in H₂O.

p. 2: “if it were produced it would rapidly reduce any available species such as an organic molecule or even protons produced by water autoionization” It seems that recombination with OH* would be the most likely outcome. Why is this not mentioned?

We did not originally mention this since it is the backward process in the reaction in Eq. (1) and hence would just modify the apparent rate of the forward reaction, i.e. the equilibrium arrows indicate that the process is, of course, possible. For extra clarity, the sentence now reads:

The solvated electron is highly reactive, so if it were produced it would rapidly reduce any available species such as an organic molecule or even protons produced by water autoionization, beyond the recombination with OH* via the backwards reaction in Eq. 1.

p. 3: “an explanation for the experimentally observed redox chemistry” remind readers what experiments you are referring to. Presumably H₂O₂ formation?

This sentence does need clarification and now reads:

...thus providing an explanation for the experimentally observed redox chemistry such as the production of H₂O₂.

p. 3: “Supplementary Fig. ??”

Thank you this has been corrected.

Check abstract and main text and use VEA or EA consistently. Or is there a difference? Not clear.

We use VEA throughout the entire paper now. Just referring to the electron affinity is technically ambiguous since the vertical electron affinity and adiabatic electron affinity are distinct quantities. We only deal with the vertical electron affinity here which does not consider the solvent relaxation on attachment of the electron.

p. 3: “For redox reactions to occur in water the oxidation reactions typically require the presence of OH• while the reduction reactions clearly depend on the ease of electron transfer, which in the bulk phase are limited by an unfavorable free energy.” Unclear how readers are to deal with such sweeping opening

statements. Under many conditions, the *OH will most likely abstract a hydrogen from somewhere (i.e. there will be no electron transfer). This is even true for pure water [Codorniu-Hernandez, E., Kusalik, P.G.: Mobility Mechanism of Hydroxyl Radicals in Aqueous Solution via Hydrogen Transfer. J. Am. Chem. Soc. 134, 532-538 (2012)]

This work describes the situation that OH radicals are formed by detaching an electron from a hydroxide ion. Therefore, there most certainly could be electron transfer even when the hydroxyl radical abstracts a hydrogen from elsewhere in the system. This is consistent with our discussion of experiments that observe simultaneous reduction and oxidation of organic species in the first paragraph of the introduction.

We have revised the discussion as follows:

For redox reactions to occur in water the oxidation reactions typically require the presence of OH• while the reduction reactions clearly depend on the ease of electron transfer, both of which are limited by an unfavorable free energy and possible competing reactions, but would motivate why experiments observe simultaneous reduction and oxidation of organic species.

Table 1 layout is awkward, as Num H+ seems to be directly linked to Expt

We have changed the layout to avoid this problem.

The Table 1 data of hydration enthalpies are based on 4 nm droplets (but readers only find this information after digging through the Methods).

The table caption now mentions the size of the simulated droplets.

What is the positioning of OH- and H+ in these droplets? Can the authors demonstrate that their placement of these ions is appropriate and fully equilibrated? After all, positioning of H+ and OH- in aqueous droplets is a non-trivial problem, see work by Greg Voth and others. For droplets containing multiple ions, some may be in the interior and others will be on the surface.

ReaxFF/CGeM preferentially partitions hydronium to the interface while hydroxide can be found near the surface and in the bulk region of the nanodroplet. These observations are in agreement with many other models and common interpretations of SFG experiments.

The revised text states:

ReaxFF/C-Gem has been extensively validated against experiment for pure water properties(44) and surface tension (see Methods), proton hopping mechanisms in bulk water and in reverse micelles,(46,47), and relevant here the correct partitioning of H+ and OH- to surface and bulk regions, respectively.(48–50)

Yet, the tiny error bars in Table 1 suggest that all ions share the same enthalpies. Why? Droplet structures and ion positions have to be discussed in much greater detail, before readers can be convinced that the Table 1 energetics are reliable.

There is certainly spread in the calculated hydration enthalpy values, but the standard deviation of that distribution is not what is reported in the table. This is because the experimentally determined hydration enthalpy is an ensemble averaged quantity and is therefore the mean of an underlying distribution. We note in the table caption that the uncertainties are bootstrapped standard errors. This is a standard technique for estimating the uncertainty in the mean

of a distribution. The mean of the distribution of hydration enthalpies is exactly the quantity we wish to compute and compare with experiment as is highlighted in Table 1.

p. 5: “the Rayleigh limit occurs at $q_{\max} \approx 32$ based on the surface tension of 72 mN/m for water.” How well does that water model used here reproduce this experimental surface tension? Many water models perform quite poorly. How does this aspect affect the validity of the data reported here? [Vega, C., de Miguel, E.: Surface tension of the most popular models of water by using the test-area simulation method. J. Chem. Phys. 126, 154707 (2007)]

In order to address this point, we have computed the surface tension of this model using a slab geometry. We used slabs of 512 and 1024 water molecules following the same procedure reported elsewhere (Muniz, M. C., Gartner, T. E., Riera, M., Knight, C., Yue, S., Paesani, F., & Panagiotopoulos, A. Z. (2021). Vapor–liquid equilibrium of water with the MB-pol many-body potential. The Journal of Chemical Physics, 154(21).). We find the surface tensions to be 60 mN/m without long-ranged corrections and ~ 63 mN/m with an analytical correction proposed by [Vega, C., de Miguel, E.: Surface tension of the most popular models of water by using the test-area simulation method. J. Chem. Phys. 126, 154707 (2007)] compared to 72 mN/m from experiment. Based on the reference provided by the reviewer, the ReaxFF/C-Gem model yields surface tension values that is the same as the best reported in that paper

We have added the following text to Main and Methods.

To further validate the ReaxFF/CGem model, we have computed its surface tension using 512 and 1024 water molecules in a slab geometry with the computational procedure reported by Muniz and co-workers.⁽⁸⁰⁾ We find the computed surface tensions to be 60 mN/m without long-ranged corrections, and 64 mN/m with the analytical correction proposed by Vega⁽⁸¹⁾, in reasonable agreement with the 72 mN/m from experiment and for fixed charge as well as polarizable models of water.^(81–83)

p. 6: “Fig. 2a, and illustrate the important point that electrons are much more weakly bound to OH⁻ in a charged environment.” Lots of open questions here. Any electron transfer will be associated with major changes in the hydration environment, with Eigen vs. Zundel and other hydration motifs. None of this is discussed in this work. I guess the authors try to get around this by using “vertical” energies, but it is not clear in how far this vertical approach yields data that are reliable and relevant.

The calculation of VIEs (as opposed to adiabatic ionization energies) is justified by the fact that VIEs are what is experimentally measurable. Hence our excellent agreement with experiment in the single ion case provides evidence to support our methodology. VAEs are much more difficult to measure experimentally, but are still useful for giving a sense of the effect that charge has on the stability of hydronium. Ultimately, Figures 2a and 2b are meant to illustrate that excess charge dramatically destabilizes ions in solution and then Figure 2c shows that destabilization is coulombic in nature.

VIEs and VAEs are independent of the thermodynamic arguments made in this paper, and yet we are to reproduce experimental hydration enthalpies with the same methodology for single ions as reported in Table 1. In essence we are calculating thermodynamic and energetic quantities and not on proton hopping mechanism, (although we have used ReaxFF/CGem to calculate Eigen, Zundel proton hopping mechanisms etc as stated above.)

On a related note, for a journal with a general audience, the concept of “vertical” (and its appropriateness for the calculations of this work) has to be explained much more clearly.

We have added an additional paragraph at the end of the methods section to address this comment:

Finally, it should be noted that we are computing vertical quantities, as opposed to adiabatic ones, which means the VIE and VEA calculations do not account for solvent relaxation upon detachment and attachment of electrons, respectively. This removes the need to model the dynamics of solvated radicals. Additionally, VIEs are experimentally measurable so we can use the VIE calculations to confirm the reliability of our methodology (*a posteriori*). Note that our hydration enthalpy calculations do, of course, account for solvent response to the presence of an ion.

p. 9: “It has been observed that many redox reactions are accelerated in aqueous microdroplets, ... The observed oxidation typically occurs via OH•” AND “In summary, our work helps explain why there are many organic redox reactions that are accelerated in microdroplets^{6,10,11}” This is a massive oversimplification. The authors have to tone down their claims. While *OH may be involved in some types of droplet acceleration, the favorability of *OH formation seen here will only apply to a small sub-set of these reactions. There are many droplet redox processes that do not involve *OH [see for example Scheme 1 in Chen, C.J., Williams, E.R.: The role of analyte concentration in accelerated reaction rates in evaporating droplets. Chem. Sci. 14, 4704-4713 (2023)].

We have changed the quoted text to be less all encompassing:

It has been observed that many redox reactions are accelerated in aqueous microdroplets, as well as many organic reactions that are found to occur in droplets but do not occur in the bulk liquid.^(6,10,11) In this work we consider a class of reactions in which the observed oxidation typically occurs via OH• and the observed reductions clearly depend on electron transfer, which we have shown becomes thermodynamically favorable in charged droplets well below the Rayleigh limit that is relevant to hydrogen peroxide formation for example.

In summary, our work provides a thermodynamic explanation for why many organic redox reactions are accelerated in microdroplets, especially those that result in the addition of hydroxyl radicals or formation of anion radicals^(6,10,11).

We hope this clarification is sufficient. This is because the conclusions of this paper truly are relevant to many organic redox reactions occurring in microdroplets. The paper cited by the reviewer describes a bimolecular redox reaction. The rate acceleration in that paper is attributed to concentration enhancements which are known to be relevant to rate accelerations in sprayed droplets (Wei, Z., Li, Y., Cooks, R. G., & Yan, X. (2020). Accelerated reaction kinetics in microdroplets: Overview and recent developments. Annual Review of Physical Chemistry, 71, 31-51.). Some papers (which we cite) have observed unimolecular redox reactions that occur in microdroplets, which strongly suggests that the solvent is involved in the reaction. Our work is most relevant to those reactions. Our work also provides a mechanism for the production of H₂O₂ in microdroplets. For these reasons, we think the strength of our conclusions is warranted.

Figure 1 is highly speculative. It seems to suggest that droplet fission is an essential component of the model proposed in this work. Why would process (a) not occur in the larger droplet shown at the top? Instead of showing such speculative cartoons as the first (!) figure in this manuscript, it would be better to start off with actual data, such as droplet structures used for generating the data in Table 1. Please use additional and better figures to explain the strategy used in this work, as well as illustrating the findings.

The reviewer is correct that droplet fission is an essential component of the proposed model since droplet fission upon spraying of water is how microdroplets are formed, and different thermodynamic pathways are possible depending on fission process. In fact Figure 1 makes use of Table 1 as the source of quantitative data in the paper that informs the figure.

To address reviewer concerns, we have revised the Figure 1 caption as follows.

Figure 1: Schematic illustration of two mechanisms for redox chemistry in charged microdroplets. The ultrasonication, electrospray, as well as gas nebulization process forms microdroplets via fission processes that create microdroplets with a net charge such that redox reactions have different thermodynamic pathways that become viable. (a) According to Table 1, within a net-charged droplet containing residual quantities of the counter-ion, an excess of 12 OH^- ions ($\sim 40\%$ of the Rayleigh limit) or 16 H^+ ions ($\sim 50\%$ of the Rayleigh limit) are sufficient to overcome the thermodynamic barrier described in Eq. 3. (b) According to Table 1, two charged droplets that are near enough so that electrons are transferred from the negative to positive droplet requires an excess of 8 OH^- ions and 8 H^+ ions ($\sim 20\text{-}25\%$ of the Rayleigh limit) to overcome the thermodynamic barrier.

Reviewer #1 (Remarks to the Author):

The authors state: "At present, however, the molecular origins of the strong redox properties of microdroplets are unclear." I believe that the situation is better than that in that OH has been conclusively identified in microdroplets in several publications and that implies that it arises from the loss of an electron from OH⁻. That all possible ways this can happen have not been determined is correct but the molecular origins of strong redox properties to me clearly point to the presence of H and OH at the water microdroplet interface.

Reviewer #2 (Remarks to the Author):

In the resubmission and revised manuscript, the authors addresses the questions that I raised previously.

My first question were addressed successfully in this round. They claimed that they used the flexible model which is able to describe the proton hopping processes in bulk water. But I still believe that a full ab initio description will be important for the chemistry here.

The second question was addressed partially. The authors gave a convincing argument on the observed difference between OH⁻ and H⁺. On the other hand, the dielectric constant discussed here (1.0 and 1.3) are the screening on the electronic process without molecular responses. Therefore, 1.0(H⁺) and 1.3(OH⁻) should be more appropriately compared with ~1.8 (bulk water). Indeed, because of the solvation structure of OH⁻ on both surface and inside the droplet, 1.3 (OH⁻) is indeed close to 1.8 (bulk water), which is very reasonable. However, I am now sure if the "dielectric saturation" is an appropriate theory to be applied here? This is because original "dielectric saturation" theory by Debye in 1940s was mainly dealing with molecular response of dielectric constant (classical E&M, at low frequency). I believe that the screening of VIE and VEA were mostly from the purely electronic part (~1.8 for bulk water), right? I hope authors can further clarify the physics here.

The question (3) on typos were successfully addressed.

Reviewer #3 (Remarks to the Author):

The authors have addressed most of the reviewers' comments. The paper can now be accepted as is.

Reviewer #1: The authors state: "At present, however, the molecular origins of the strong redox properties of microdroplets are unclear." I believe that the situation is better than that in that OH has been conclusively identified in microdroplets in several publications and that implies that it arises from the loss of an electron from OH⁻. That all possible ways this can happen have not been determined is correct but the molecular origins of strong redox properties to me clearly point to the presence of H and OH at the water microdroplet interface

We do understand the reviewer's position. We believe our role is to develop independent analysis and theory for understanding the energetics of the process of creating hydroxy radicals, to help support (or refute!) the position of Reviewer 1.

Reviewer #2: My first question were addressed successfully in this round. They claimed that they used the flexible model which is able to describe the proton hopping processes in bulk water. But I still believe that a full ab initio description will be important for the chemistry here.

Full ab initio sampling is not possible, but is based on a reactive force field that can sample. We believe our high quality ab initio model on top of configurations has been shown to be quantitative.

The second question was addressed partially. The authors gave a convincing argument on the observed difference between OH⁻ and H⁺. On the other hand, the dielectric constant discussed here (1.0 and 1.3) are the screening on the electronic process without molecular responses. Therefore, 1.0(H⁺) and 1.3(OH⁻) should be more appropriately compared with ~1.8 (bulk water). Indeed, because of the solvation structure of OH⁻ on both surface and inside the droplet, 1.3 (OH⁻) is indeed close to 1.8 (bulk water), which is very reasonable. However, I am now sure if the "dielectric saturation" is an appropriate theory to be applied here? This is because original "dielectric saturation" theory by Debye in 1940s was mainly dealing with molecular response of dielectric constant (classical E&M, at low frequency). I believe that the screening of VIE and VEA were mostly from the purely electronic part (~1.8 for bulk water), right? I hope authors can further clarify the physics here.

We thank the reviewer for these important comments, and hope the reviewer appreciates the following physical picture.

First we understand that the reviewer concludes that we should compare against the optical dielectric constant of ~1 which is dielectric screening from the electronic component, but not the screening due to dipole reorientation (~78) since we did not include the molecular response after *the neutral radical* is formed.

But there are two molecular responses – how water dipoles reorient to ions (OH⁻ or H⁺) and how they reorient to the neutral radicals. The molecular response is much larger for ions compared to neutral species. We argue that the molecular response to the ions was accounted for, because the orientation of water around OH⁻ ions from the MD simulations encodes the dielectric screening, i.e. the embedded charges used in the ab initio calculations are arranged such that they poorly screen the ions. Hence we interpret the values of 1 and 1.3 as a drop from 80 due to dielectric saturation. This is consistent with the fact that the 2 OH⁻ calculations show a VIE which is shifted less than one would expect if we referenced the optical dielectric constant of ~1, since the embedded charges did provide enough screening that is consistent with a dielectric constant much larger than 1 (i.e. close to 80). The reviewer is right that we did not describe the response to the neutral radical, but this can be neglected as the ion response is the dominant effect.

We have added the following text to the manuscript:

We note that the dielectric constant that emerges of 1.0 and 1.3 takes into account not only the screening on the electronic process (embodied in the optical dielectric constant of 1.77), but also implicitly includes the molecular response of dipole reorientation around the OH⁻ and H⁺ ions (which dominates the static dielectric constant of ~78). This is because the embedded charges from the MD used in the ab initio calculations come from water arrangements that reflect the dipole orientation response around the ions. This is consistent with the fact that for the 2 OH⁻ calculations the VIE is shifted less than one would expect if we referenced the optical dielectric constant of ~1, since the embedded charges provided screening that is more consistent with a dielectric constant of 80. But starting with 4 OH⁻ or 4 H⁺ ions, the shifts in the VIE and VEA with increasing droplet charge imply that there is a rapid onset of dielectric saturation, in which the static dielectric constant of water decreases because water molecules in the presence of ions cannot rearrange

their dipoles to screen out large numbers of ionic charges. \cite{babu1999theory,gavish2016dependence} While our calculations do not describe the water response to the neutral radical, this can be neglected as the water response to multiple ions is the dominant effect, and is accounted for in our theory.”

Reviewer #3 (Remarks to the Author): The authors have addressed most of the reviewers' comments. The paper can now be accepted as is.

We are glad that Reviewer 3 is satisfied by the previous changes and their support for publication.

Reviewer #2 (Remarks to the Author):

In the current version, I feel that the authors have successfully addressed all my previous questions. I would like to recommend its publication, but I have several final comments and suggestions for the authors for which I think they might be important to further improve the quality of the work.

Indeed, it is important that the authors have clarified that the dielectric screening ~ 1 was in comparison with the static dielectric constant of ~ 80 liquid water. Since the snapshots were taken from molecular dynamics simulation at finite temperature, the ionic (molecular) screening is naturally considered.

At first glance, the dielectric constant of about 1 is indeed very small compared to 80 in bulk water. However, considering the nano-size water droplets modeled in this work, the predicted or inferred low dielectric constant here is not too surprising. Recently, there are lots of calculations as well as experiments that have been made in nanoconfined water, which is very similar to the nanoscale water droplets modeled in this work. Although there are debates on some detailed physical effects, consensus has been made on the major origin of the significantly reduced static dielectric constant which was due to the loss of dipolar correlations in confined water, please refer to (Phys. Rev. Lett., 117, 048001 2016 and J. Phys. Chem. Lett. 12, 4319 2021) I think similar physics could appear here, since nanoscale water droplets have vacuum-water interfaces that disrupt the tetrahedral hydrogen bond network in bulk water, therefore disrupting the dipole correlations. In this regard, there is more recent progress in the field concerning Debye's dielectric saturation theory (Phys. Rev. Lett. 131, 076801 2023). In the above work, a more modernized picture of "dielectric saturation" picture has been given in terms of loss of dipole correlation by the intrusion of hydration shell instead of the effect of high electric field in the conventional Debye's hypothesis in 1920s. Considering the above important progresses in the field, the authors might want the above references and at the same time have minor revisions on the discussion of dielectric saturation which should include the effect of interrupted dipolar correlations in the reduced dielectric constant.

Reviewer #2 (Remarks to the Author): In the current version, I feel that the authors have successfully addressed all my previous questions. I would like to recommend its publication, but I have several final comments and suggestions for the authors for which I think they might be important to further improve the quality of the work.

Indeed, it is important that the authors have clarified that the dielectric screening ~ 1 was in comparison with the static dielectric constant of ~ 80 liquid water. Since the snapshots were taken from molecular dynamics simulation at finite temperature, the ionic (molecular) screening is naturally considered.

At first glance, the dielectric constant of about 1 is indeed very small compared to 80 in bulk water. However, considering the nano-size water droplets modeled in this work, the predicted or inferred low dielectric constant here is not too surprising. Recently, there are lots of calculations as well as experiments that have been made in nanoconfined water, which is very similar to the nanoscale water droplets modeled in this work. Although there are debates on some detailed physical effects, consensus has been made on the major origin of the significantly reduced static dielectric constant which was due to the loss of dipolar correlations in confined water, please refer to (Phys. Rev. Lett., 117, 048001 2016 and J. Phys. Chem. Lett. 12, 4319 2021) I think similar physics could appear here, since nanoscale water droplets have vacuum-water interfaces that disrupt the tetrahedral hydrogen bond network in bulk water, therefore disrupting the dipole correlations. In this regard, there is more recent progress in the field concerning Debye's dielectric saturation theory (Phys. Rev. Lett. 131, 076801 2023). In the above work, a more modernized picture of "dielectric saturation" picture has been given in terms of loss of dipole correlation by the intrusion of hydration shell instead of the effect of high electric field in the conventional Debye's hypothesis in 1920s. Considering the above important progresses in the field, the authors might want the above references and at the same time have minor revisions on the discussion of dielectric saturation which should include the effect of interrupted dipolar correlations in the reduced dielectric constant.

The reviewer makes an interesting observation that water under nano-confinement also results in a diminished dielectric constant. The provided references are useful. As such, we have added the following text to our discussion of the relevant section:

"Additionally, nano-confined water has been shown to develop an asymmetric dielectric profile where the dielectric constant perpendicular to a surface is decreased significantly. \cite{schlaich2016water,olivieri2021confined} This effect and more modern literature attribute dielectric saturation as arising from the loss of dipolar correlations in water. For example in the presence of salt the water hydrogen-bond network is disrupted by the presence of hydration shells of the ions, \cite{zhang2023dissolving} which in turns suppresses the collective dielectric response. This situation is similar to our net charged droplets in which the overall dipole fluctuations are dominated by the rearrangement of net charge in the droplet, and further exacerbated by the presence of an interface that also breaks up the dipole correlations observed in the bulk solvent."

We thank the reviewer for their helpful and interesting comments.